# LMEXPLAINER: A KNOWLEDGE-ENHANCED EXPLAINER FOR LANGUAGE MODELS

## ABSTRACT

Language models (LMs), such as GPT-4, are powerful tools for natural language processing, capable of handling diverse tasks from text generation to question answering. However, their decision process lacks transparency due to the complex, multi-layered, and nonlinear model structures involving millions of parameters. This hinders user trust on LMs, especially in safety-critical applications. Due to the opaque nature of LMs, a promising approach for explaining how they work is by generating explanations on a more transparent surrogate (e.g., a knowledge graph (KG)). Such works mostly exploit attention weights to provide explanations for LM recommendations. However, pure attention-based explanations lack scalability to keep up with the growing complexity of LMs. To bridge this important gap, we propose LMExplainer, a knowledge-enhanced explainer for LMs capable of providing human-understandable explanations. It is designed to efficiently locate the most relevant knowledge within a large-scale KG via the graph attention neural network (GAT) to extract key decision signals reflecting how a given LM works. Extensive experiments comparing LMExplainer against seven state-of-the-art baselines show that it outperforms existing LM+KG methods on the CommonsenseQA and OpenBookQA datasets. We compare the explanation generated by LMExplainer with other algorithm-generated explanations as well as human-annotated explanations. The results show that LMExplainer generates more comprehensive and clearer explanations.

## 1 INTRODUCTION

Pre-trained language models (LMs) have recently attracted significant attention due to their impressive state-of-the-art (SOTA) performance on various natural language processing (NLP) tasks (Brown et al., 2020; Liu et al., 2023; Wei et al.; Zhou et al., 2022; Li et al., 2022). These tasks include language translation (Conneau & Lample, 2019), text generation (Mireshghallah et al., 2022), and text classification (Raffel et al., 2020), among others. One of the main advantages of LMs is their ability to capture the nuances and the complexity of human languages.

However, a major limitation of LMs is a lack of interpretability (Meng et al., 2022). It is often difficult to provide explanations about their "black box" decision-making processes. LMs use techniques such as attention mechanisms, which allow them to focus on specific parts of the input data when making decisions (Vaswani et al., 2017; Devlin et al., 2019; Liu et al., 2019a). These mechanisms can be difficult for people to understand, as they produce abstract and non-transparent internal learning representations (Jain & Wallace, 2019). For example, a model embedding might capture relationships and meanings as a result of passages through millions of neurons. However, such meanings might not be immediately apparent to humans. This lack of interpretability poses a challenge to mission critical domains (e.g., healthcare (Loh et al., 2022) and online education (Zytek et al., 2022)) as it hampers users' trust on the recommendations made by the models.

Due to the opaque nature of LMs, a promising approach for explaining how they work is by generating explanations on a more transparent surrogate (e.g., a knowledge graph (KG)). (Geng et al., 2022) leverages a KG as a submodel to enhance the explainability of LM-based recommendations. Such methods provide insights into how to interpret the complex model by translating it into more comprehensible counterparts. Attention-based explanations have also gained significant attention. For instance, (Vig, 2019) proposes a visualizing method for attention in the LM, enhancing our

understanding of how these models allocate focus across input tokens. However, (Zini & Awad, 2022) pointed out that attention is not equal to explanation. Individual token representations are not enough. A surrogate that maps tokens to specific knowledge elements that align with the reasoning process of the LM is imperative.

In this paper, we explore the potential of using explanations to serve two purposes: 1) helping humans in understanding the model, and 2) enhancing the model's understanding of the task at hand through interpretation during the explanation process. In this paper, explanation refers to explaining the model's decision-making in a human-understandable way, while interpretation refers to understanding the internal workings of the model. To address the limitations of current approaches, we propose the `LMExplainer` approach. It is a novel method for explaining the recommendations made by LMs. It is designed to efficiently locate the most relevant knowledge within a large-scale KG via the graph attention neural network (GAT) (Veličković et al., 2018) to extract key decision signals reflecting the rationale behind the recommendations made by LMs.

We experimentally evaluate `LMExplainer` on the question-answering (QA) task using the CommonsenseQA (Talmor et al., 2019) and OpenBookQA (Mihaylov et al., 2018) datasets. The results demonstrate that `LMExplainer` outperforms SOTA LM+KG QA methods on CommonsenseQA, while achieving comparable performance on OpenBookQA. Furthermore, we demonstrate that `LMExplainer` is capable of providing useful insights on the reasoning processes of LMs in a human understandable form, surpassing prior explanation methods. To the best of our knowledge, `LMExplainer` is the first work capable of leveraging graph-based knowledge in generating natural language explanations on the rationale behind LM behaviors.

## 2 RELATED WORK

Post-hoc explanation methods have attracted significant attention in NLP research in recent years. Ribeiro et al. proposed LIME, which generates explanations by approximating the original model with a local sample and highlights the most important features. Guidotti et al. extended it with a decision tree classifier to approximate deep models. However, they cannot guarantee that the approximations are accurate representations of the original model due to inherent limitations of decision trees. Thorne et al. generate concepts of classifiers operating on pairs of sentences, while Yu et al. generate *aspects* as explanations for search results. Kumar & Talukdar used positive labels to generate candidate explanations, while Chen et al. used contrastive examples in the form of "why A not B" to distinguish between confusing candidates. Different from prior work, we integrate reasoning features and concepts into `LMExplainer` to explain LM behaviors.

Recently, language models (LMs) such as RoBERTa (Liu et al., 2019a) and GPT-4 (OpenAI, 2023) have achieved impressive results. However, these models lack interpretability, which can hinder their adoption in mission critical real-world applications. Previous interpretable frameworks (Ribeiro et al., 2016; Sundararajan et al., 2017; Smilkov et al., 2017; Ding & Koehn, 2021; Swamy et al., 2021) can be applied to LMs. However, they often rely on approximations and simplifications of the original models, which can result in discrepancies between the model behaviours and the explanations. In contrast, `LMExplainer` explains LMs by illustrating the model reasoning process.

KGs are increasingly adopted as a means to improve the interpretability and explainability of LMs (Huang et al., 2022; Yasunaga et al., 2021; Huang et al., 2019; Liu et al., 2019b). KGs are structured representations of knowledge, and can be used to capture complex semantic relationships that are difficult to represent in traditional LMs (Ji et al., 2021). Zhan et al. (2022a) retrieves explainable reasoning paths from a KG and uses path features to predict the answers. Yasunaga et al. (2021) integrates the KG into the model, enabling the model to reason over structured knowledge and generate more interpretable predictions. However, these explanations can be inconsistent and accurate representations of the model reasoning process. In addition, they are difficult for humans to understand as they are being represented in a graph-based format. By drawing upon insights from prior works, `LMExplainer` employs graph embedding to generate explanations to address these limitations.

## 3 THE PROPOSED LMExplainer APPROACH

The LMExplainer architecture is shown in Figure 1. It consists of three main steps: **(1) key element extraction and building** (Section 3.2), **(2) element-graph interpretation** (Section 3.3), and **(3) explanation generation** (Section 3.4). In the first step, we extract the relevant elements from the input data and the knowledge retrieved from the KG, and build an element-graph representation. In the second step, we leverage GAT to interpret the element-graph and identify the *reason-elements* behind LM predictions. In the third step, we design an instruction-based method to generate human-understandable explanations of the decision-making process based on the identified *reason-elements*. LMExplainer is flexible and applicable to a range of LMs (e.g., RoBERTa (Liu et al., 2019a), GPT-2 (Radford et al.), and Llama (Touvron et al., 2023)).

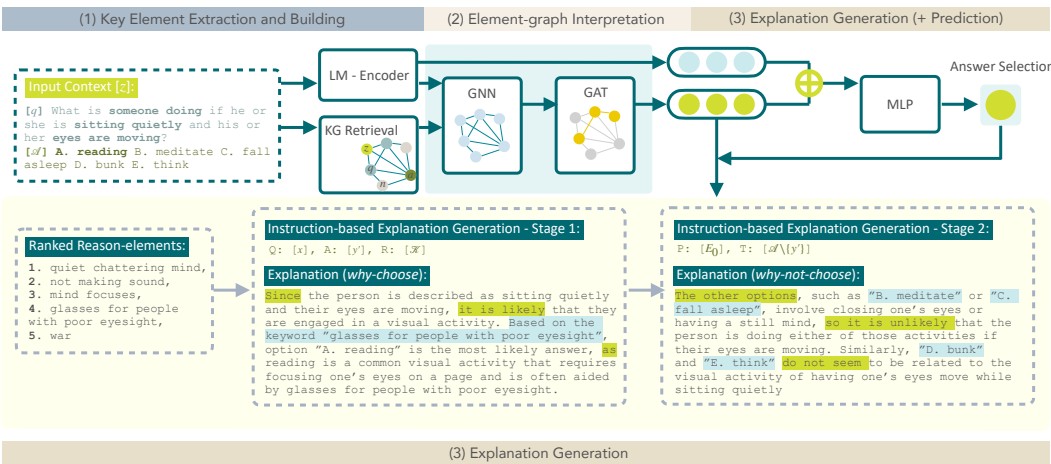

Figure 1: The LMExplainer architecture. Given an input content $z$, we first generate language embeddings using a pre-trained LM. Simultaneously, it retrieves relevant knowledge from a KG to construct a subgraph. The language embeddings and subgraph are then combined to obtain GNN embeddings. This combined representation is then passed through a GAT to obtain the attention. The attention serves two purposes. Firstly, it weighs the importance of the GNN embeddings and is used with the language embeddings for the final prediction. Secondly, they are used to generate explanations by highlighting the most important parts of the reasoning process.

### 3.1 TASK DEFINITION

We define the task of generating reasoning-level explanations for inferences made by LMs. As an example, we use a QA task. Given a pre-trained LM $f_{LM}$ with input question $q$, answer choice set $\mathcal{A}$ and predicted answer $y' \in \mathcal{A}$, the goal is to generate an explanation $E_0$ for why $f_{LM}$ chooses $y'$ and an explanation $E_1$ for why $f_{LM}$ does not choose other options $\mathcal{A} \setminus \{y'\}$. This task can be expressed as:

$$(E_0, E_1) \leftarrow GenerateExplanation(f_{LM}, q, \mathcal{A}, y'). \quad (1)$$

### 3.2 KEY ELEMENTS EXTRACTION AND BUILDING

Certain key elements can significantly influence the reasoning process of LMs. To capture these essential elements, we first tokenize a set of sentences $\{q\} \cup \mathcal{A}$ into tokens $\{x_1, x_2, \ldots, x_n\}$. Let $z$ denote this set of resulting tokens. Figure 1 illustrates the "Input Content $[z]$". The tokens $z$ are then used to construct a multi-relational graph, following the approach from Yasunaga et al.. Firstly, the $L$-hop neighbor $G_k$ of $z$ is extracted from ConceptNet (Speer et al., 2017) to integrate external knowledge, following the approach from Feng et al. (2020). However, $G_k$ can still contain a large number of edges, which lead to a huge reasoning space. Our main goal is therefore to construct a relevant sub-graph of $G_k$, referred to as the *element-graph* $G_e$. This allows us to identify essential elements that play a key role, and analyze the relations among them. We integrate the embedding from LMs to guide the pruning for $G_k$. Specifically, for every node $v$ in $G_k$, we define an associated

score for pruning purposes, which is expressed as:

$$v_{score} = f_{prob}(z_{emb}, v_{emb}), \tag{2}$$

where $f_{prob}$ is the probability computation function of the pre-trained LM, $z_{emb}$ and $v_{emb}$ are the embeddings derived from textual representations of $z$ and $v$ respectively, are concatenated to $f_{prob}$. The score captures the correlation between the node $v$ and input content $z$, and is used to remove irrelevant nodes. We select the top $K$ nodes based on their scores. The resulting pruned graph is denoted by $G_e$, which is referred to as the *element-graph*. We outline the procedure for constructing the *element-graph* in Algorithm 1.

---

**Algorithm 1:** Construct Element-graph

**Data:** Input content $z$
**Result:** Pruned element-graph $G_e$
1 **begin**
2      $G_k \leftarrow$ ExtractFromConceptNet($z$)
                     `// Extract the` $L$`-hop neighbor from ConceptNet`
3      **for** *each node $v_e$ in $G_k$* **do**
4          $v_{score} \leftarrow f_{prob}(z_{emb}, v_{emb})$                `// Compute score for pruning`
5      **end**
6      $G_e \leftarrow$ SelectTopK($G_k$)                  `// Prune based on top` $K$ `scores`
7      **return** $G_e$
8 **end**

---

### 3.3 ELEMENT-GRAPH INTERPRETATION

Given an element-graph $G_e$, we follow Yasunaga et al. (2021) to extract the representation for graph reasoning. The method leverages the GAT (Veličković et al., 2018) to preserve the structure and context of the input through the connections between the nodes. Veličković et al. use the graph attention operation to take a set of node features as input and output a corresponding set of new node features. Formally, the input to the $k$th attention layer is denoted as $\boldsymbol{h}_k = \{h_{k1}, h_{k2}, \ldots, h_{kN}\}$, where $h_{kj} \in \mathbb{R}^F$ is the intermediate feature for node $v_j$, $F$ is the input feature size and $N$ is the number of nodes in the graph. The attention layer outputs a new set of corresponding node features, $\boldsymbol{h}_{k+1} = \{h_{k+1,1}, h_{k+1,2}, \ldots, h_{k+1,N}\}$ with $h_{kj} \in \mathbb{R}^{F'}$.

A parameterized transformation $m : \mathbb{R}^F \to \mathbb{R}^M$ is first applied to $\boldsymbol{h}_k$ to generate the transformation $m(\boldsymbol{h}_k)$. A parameterized self-attention mechanism $a : \mathbb{R}^F \times \mathbb{R}^F \to \mathbb{R}$ is then used to obtain attention scores on $\boldsymbol{h}_k$. To retain structural information within the graph, attention scope for node $v_i$ is limited to nodes in its 1-hop neighborhood which is denoted as $\mathcal{N}_i$. Furthermore, the attention scores are normalized over the neighborhood $\mathcal{N}_i$ to generate attention coefficients:

$$\alpha_{ij} = \frac{\exp(a(h_{ki}, h_{kj}))}{\sum_{v_l \in \mathcal{N}_i} \exp(a(h_{ki}, h_{kl}))}. \tag{3}$$

The output feature $h_{k+1,i}$ is an attentive linear combination of neighboring features with an optional activation:

$$h_{k+1,i} = \sigma\left(\sum_{v_j \in \mathcal{N}_i} \alpha_{ij} m(h_{kj})\right) \tag{4}$$

We build the graph reasoning network based on the above graph attention operation. Specifically, we employ a parameterized MLP $f_m$ for feature transformation. This MLP $f_m$ explicitly associates the node $v_i$ with its neighboring nodes $v_j \in \mathcal{N}_i$ by processing the feature $h_{ki}$, the recorded node type $u_i$ of node $v_i$ and the recorded relation types $r_{ij}$ to $v_j$, all of which are sourced from the element-graph. The attention scores $\alpha_{ij}$ are computed using another parameterized MLP that takes features $h_{ki}, h_{kj}$, node and relation types $u_i, r_{ij}$ and node scores of $v_i$ and $v_j$ as input. The detailed information can be found in the Appendix B.2.

The output activation is implemented as a third 2-layer parameterized MLP $f_\sigma$ and the output features are thus obtained by:

$$h_{k+1,i} = f_\sigma \big( \sum_{v_j \in \mathcal{N}_i} \alpha_{ij} m(h_{kj}, u_i, r_{ij}) \big) + h_{kj}, \tag{5}$$

where the output feature size is the same as the input feature size. The initial input features $\boldsymbol{h}_0$ is obtained by a linear transformation of node embeddings $v_{emb}$.

### 3.3.1 LEARNING AND INFERENCE

In our task, each question $q$ is associated with a set of answer choices $\mathcal{A}$, with only one being the correct answer. We leverage the information from the LM embedding and the node embedding from the element-graph. Specifically, we define the probability of choosing an answer as $P(a|q) \propto \exp(MLP(\mathbb{H}^{LM}, \boldsymbol{h}_K, \boldsymbol{\alpha}_K))$, where $\boldsymbol{h}_K$ is the output features and $\boldsymbol{\alpha}_K$ is the last-layer attention coefficients of a K-layer graph reasoning network given $G_e$ as input, and $\mathbb{H}^{LM}$ is the representation embedding from LM. The corresponding nodes (i.e., the *reason-elements*) in $G_e$ are used to generate textual explanations about the decision-making process of the LM. We optimize the model by minimizing the cross-entropy loss.

---

**Algorithm 2:** Element-graph Interpretation

**Data:** Element-graph $G_e$ containing node type embedding $u_i$ and relation embedding $r_{ij}$, input $z$.

**Result:** *Reason-elements*

1  **begin**
2      **for** *each attention layer $k$ in graph reasoning network* **do**
3          **for** *each node $v_i$ in $G_e$* **do**
4              $\alpha_{ij} \leftarrow \frac{\exp(a(h_{ki}, h_{kj}, u_i, r_{ij}))}{\sum_{v_l \in \mathcal{N}_i} \exp(a(h_{ki}, h_{kl}))}$       // Compute attention coefficient $\alpha_{ij}$
5              $h_{k+1,i} \leftarrow f_\delta \left( \sum_{v_j \in \mathcal{N}_i} \alpha_{ij} m(h_{kj}, u_i, r_{ij}) \right) + h_{ki}$     // Update node feature
6          **end**
7      **end**
8      $\mathbb{H}^{LM} \leftarrow f_{LM}(z)$                            // Forming $\mathbb{H}^{LM}$
9      $P(a|q) \propto \exp(MLP(\mathbb{H}^{LM}, \boldsymbol{h}_K, \boldsymbol{\alpha}_K))$     // Probability of choosing an answer
10     ReasonElements $\leftarrow$ RankNode($G_e, \boldsymbol{\alpha}_K$))     // Rank nodes based on the attentions
11     **return** *ReasonElements*
12 **end**

---

### 3.4 ATTENTION-AWARE EXPLANATION GENERATION

In prior work, Chen et al. (2021) proposed a counterfactual explanation generator that pairs input text with counterfactual examples to fine-tune LMs to generate explanations in the form of "why A and not B". However, this approach does not provide a complete explanation of the LM decision-making process. In contrast, the LMExplainer explanation generator consists of two steps: 1) explanation component extraction, and 2) instruction-based explanation generation.

### 3.4.1 EXPLANATION COMPONENT EXTRACTION

We first extract the key components that are essential to the LM decision-making process. These key components consist of the final answer, *reason-elements* and the attention $\alpha$. The final answer and *reason-elements* are used to trace the important explanation nodes. The attention is used to sort the nodes and select the top $w$ nodes most relevant to the decision. Each node represents an element, so we have $w$ most important components for the explanation. We use $\mathcal{K}$ to represent the set of extracted key components. The output, $E$, is a natural language explanation. We outline the procedure to interpret the *element-graph* and extract the *reason-elements* in Algorithm 2.

### 3.4.2 INSTRUCTION-BASED EXPLANATION GENERATION

We integrate the key component set $\mathcal{K}$ into our instruction-based explanation generator. To guide the generation of explanations, we leverage a set of predefined structures, including the input content $z$, model predicted output $y'$, the trigger sentences, and the extracted key components $\mathcal{K}$. The `LMExplainer` explanation generation involves two stages: (1) *why-choose* for explaining why the model chose the specific answer, and (2) *why-not-choose* for explaining why the model did not choose the other explanations. In the *why-choose* stage, we use instructions in the form of "Q: $[z]$, A: $[y']$, R: $[\mathcal{K}]$". The *why-choose* explanation is denoted as $E_0$. In the *why-not-choose* stage, we use instructions in the form of "P: $[E_0]$, T: $[\mathcal{A} \setminus \{y'\}]$". Q, A, R, P and T are instructions for GPT-3.5-turbo (Ouyang et al., 2022) to generate the literal explanations of the reasoning process of a given LM. The generator outputs a natural language explanation in the form of a sentence or a paragraph. The details of our instruction are shown in Appendix C.3.

## 4 EXPERIMENTAL EVALUATION

### 4.1 EXPERIMENT SETTINGS

In our experiments, we use the CommonsenseQA (Talmor et al., 2019) and OpenBookQA (Mihaylov et al., 2018) datasets to evaluate the performance of the candidate approaches. CommonsenseQA consists of 12,247 questions created by crowd-workers, which are designed to test commonsense knowledge through a 5-way multiple-choice QA task. OpenBookQA consists of 5,957 questions each requiring the task of 4-way multiple choice question answering. The questions are designed to assess the ability of models to reason with elementary science knowledge.

Our evaluation can be divided into two parts. In the first part, we focus on model performance. We compare `LMExplainer` with three sets of baseline models on the CommonsenseQA and Open-BookQA datasets. The first set of baseline models consists of fine-tuned LM RoBERTa-large (Liu et al., 2019a), which demonstrates the capabilities of language models without interpretation. The second set of baseline models includes KG-augmented versions of RoBERTa-large, using Concept-Net as the source of common sense knowledge and following the approach in (Lin et al., 2019). The third set of baseline models is the current SOTA commonsense reasoning method on Common-senseQA, MHGRN (Feng et al., 2020), QA-GNN (Yasunaga et al., 2021), GreaseLM (Zhang et al., 2022). The LMs we used are from Huggingface[1].

In the second part, we evaluate `LMExplainer` on explanation ability. To establish a baseline for comparison, two prior works, namely PathReasoner (Zhan et al., 2022a) and Explanations for Com-monsenseQA (ECQA) (Aggarwal et al., 2021), were employed as benchmarks. These works are recognized for providing natural and comprehensible explanations.

We set our GNN module to have 200 dimensions and 5 layers, where a dropout rate of 0.2 was applied to each layer. We trained the model using the RAdam optimizer on a single NVIDIA A100 GPU. A batch size of 64 was employed during the training process, and the learning rate for the language model and the GNN module were set to $1e-5$ and $1e-3$, respectively. These settings were adopted in the first part of the evaluation to investigate the performance of the GNN module.

We employ ConceptNet (Speer et al., 2017) as our external knowledge source for CommonsenseQA and OpenBookQA. ConceptNet contains a vast amount of information with 799,273 nodes and 2,487,810 edges, which provides a valuable resource for improving the accuracy of QA systems. We extract the $G_k$ with a hop size of 2, and subsequently prune the obtained graph to retain only the top 200 nodes.

### 4.2 RESULTS AND DISCUSSION

We present our experimental results in Table 1 and Table 2, where the accuracy of our proposed `LMExplainer` is evaluated on the CommonsenseQA and OpenBookQA datasets. Our empirical findings indicate that `LMExplainer` leads to consistent improvements in performance compared to existing baseline methods on both datasets. Specifically, the test performance on Common-senseQA is improved by 4.71% over the prior best LM+KG method, GreaseLM, 5.35% over the

---

[1]https://huggingface.co/

included KG augmented LMs, and 7.12% over fine-tuned LMs. The test performance achieves comparable results to the prior best LM+KG method, GreaseLM, on OpenBookQA. It is worth noting that GreaseLM is specifically designed to improve accuracy for QA tasks, while our LMExplainer focuses on providing explanations for the reasoning process. Despite this difference in focus, our LMExplainer not only offers insight into the underlying reasoning but also demonstrates an improvement in performance. This finding highlights the potential benefits of incorporating explainability into the model design, as it may lead to enhanced performance in addition to fostering a better understanding of the decision-making process.

| Method | IHdev-Acc. | IHtest-Acc. |
|---|---|---|
| **Baselines** (Feng et al., 2020) | | |
| MHGRN (2020) | 73.69% | 71.08% |
| KagNet (2019) | 73.47% | 69.01% |
| GconAttn (2019) | 72.61% | 68.59% |
| RGCN (2018) | 72.69% | 68.41% |
| RN (2017) | 74.57% | 69.08% |
| **Baselines** (our implementation) | | |
| GreaseLM (2022) | 76.17% | 72.60% |
| QA-GNN (2021) | 74.94% | 72.36% |
| LMExplainer (ours) | **77.97%** | **77.31%** |

Table 1: Performance comparison of our proposed LMExplainer model against various baselines on Commonsense QA in-house split. Our model outperforms all the other methods, achieving 77.97% and 77.31% accuracy on IHdev and IHtest, respectively. As the official test is hidden, here we report the in-house Dev (IHdev) and Test (IHtest) accuracy, following the data split of (Lin et al., 2019).

| Method | Dev-Acc. | Test-Acc. |
|---|---|---|
| **Baselines** (Feng et al., 2020) | | |
| MHGRN (2020) | 68.10% | 66.85% |
| GconAttn (2019) | 64.30% | 61.90% |
| RGCN (2018) | 64.65% | 62.45% |
| RN (2017) | 67.00% | 65.20% |
| **Baselines** (our implementation) | | |
| GreaseLM (2022) | **71.80%** | **70.80%** |
| QA-GNN (2021) | 63.00% | 59.80% |
| LMExplainer (ours) | 69.20% | 68.00% |

Table 2: Performance comparison of our proposed LMExplainer model against various baselines on OpenBookQA. Our LMExplainer model exhibits competitive performance in relation to the top-performing model, GreaseLM. It is worth noting that GreaseLM is specifically tailored to enhance accuracy for QA tasks, whereas our LMExplainer model emphasizes providing explanations for the underlying reasoning process. We use the official data splits.

### 4.3 EXPLANATION RESULTS

Our explanation results are in Table 3. To further demonstrate the effectiveness of our approach, we compare it with two SOTA methods, PathReasoner (Zhan et al., 2022b) and ECQA (Aggarwal et al., 2021). PathReasoner utilizes structured information to explain the reasoning path, while ECQA first is created by human-annotated explanations and then leverages a generation model to organize the final explanation. In Table 3, we present the inputs and results of our approach, which include ranked *reason-elements* and explanations of the reasoning process. These examples highlight the ability of LMExplainer in generating comprehensive and interpretable explanations for the LMs. More examples are shown in Appendix D.4.

| Input Questions | Q: What is someone doing if he or she is sitting quietly and his or her eyes are moving? A. reading B. meditate C. fall asleep D. bunk E. think |
|---|---|
| Label | A. reading |
| | **Results of Our Approach - LM** |
| Ranked Reason-elements | 1. quiet chattering mind, 2. not making sound, 3. mind focuses, 4. glasses for people with poor eyesight, 5. war |
| Explanation (why-choose) | Since the person is described as sitting quietly and their eyes are moving, it is likely that they are engaged in a visual activity. Based on the keyword "glasses for people with poor eyesight", option "A. reading" is the most likely answer, as reading is a common visual activity that requires focusing one's eyes on a page and is often aided by glasses for people with poor eyesight. |
| Explanation (why-not-choose) | The other options, such as "B. meditate" or "C. fall asleep", involve closing one's eyes or having a still mind, so it is unlikely that the person is doing either of those activities if their eyes are moving. Similarly, "D. bunk" and "E. think" do not seem to be related to the visual activity of having one's eyes move while sitting quietly. |
| | **Explanation of Others** |
| PathReasoner (Zhan et al., 2022a) | quietly [related to] quiet [at location] a library [used for] reading |
| ECQA (Aggarwal et al., 2021) | While meditating and sleeping, eyes don't move, eyes are closed. |

Table 3: Explanation examples of `LMExplainer`, PathReasoner and ECQA. We show the different types of explanations, including ranked *reason-elements*, *why-choose* explanations and *why-not-choose* explanations. The explanations for *why-choose*, present the model reasoning process in a logical way, while for *why-not-choose* show the model why does not choose other answers, which enhances the transparency and interpretability of the reasoning process for humans. We use green and blue to highlight the logical connectives and reasoning framework, respectively. The complete results of comparison methods are shown in Appendix (Table 7).

| Method | IHdev-Acc. | IHtest-Acc. |
|---|---|---|
| RoBERTa w/o itp | 68.63% | 64.54% |
| RoBERTa-large w/o itp | 73.05% | 71.96% |
| RoBERTa-large + itp | **77.97%** | **77.31%** |

Table 4: Ablation study on the effect of interpreting component on model accuracy.

In comparison to PathReasoner explanations, which only provide structured reasoning paths that are non-informative and require manual selection of a specific path, our proposed approach not only offers a complete reasoning path but also provides a justification for the predicted answer. As illustrated in Table 3 and Table 7, PathReasoner presents four reasoning paths, including redundant paths, making it difficult to identify the faithful reasoning path. In contrast, our method provides a clear and concise natural language explanation for the chosen answer (*why-choose* explanation), which greatly enhances the understandability and smoothness of the explanation.

The ECQA consists of human-annotated explanations that provide highly accurate descriptions of the reasoning process. However, as shown in Table 3 and Table 7, its explanations are simply a combination of positive and negative examples provided by humans. While this approach can generate high-quality explanations from a human perspective, it fails to illustrate the actual decision-making process of the model. In contrast, the explanations generated by `LMExplainer` are not a mere combination of sentences but are inferred and logically derived. `LMExplainer` provides a more comprehensive and accurate depiction of the reasoning process and improves the overall interpretability and usefulness of the generated explanations. In addition, the *why-not-choose* explanation explains why the model does not choose other answers, which gives people a better understanding of the model's reasoning process and increases the transparency of the model. These results highlight the effectiveness of quantifying the influence of tokens on determining the decision-making process and provide a literal representation of the information flow during the inference process. This is important because it allows us to understand the rationale behind the decision-making process of the LM and identify key factors that contribute to its predictions.

| LM | IHdev-Acc. | IHtest-Acc. |
|---|---|---|
| RoBERTa | 66.26% | 63.01% |
| RoBERTa-large (final) | **77.97%** | **77.31%** |

Table 5: Ablation study on the effect of LM size on model accuracy.

| Method | IHdev-Acc. | IHtest-Acc. |
|---|---|---|
| RoBERTa only | 62.65% | 60.27% |
| RoBERTa-large only | 74.28% | 70.19% |
| RoBERTa-large + external knowledge | **77.97%** | **77.31%** |

Table 6: Ablation study on the effect of knowledge component on model accuracy.

### 4.4 ABLATION STUDIES

In this section, we examine the impact of different components of `LMExplainer` on its performance. We evaluated the effects of the size of the LMs, knowledge components, and interpreting components using the CommonsenseQA IHdev and IHtest datasets. Tables 5, 6 and 4 summarize the ablation study results.

Table 5 shows the impact of the size of LM on `LMExplainer`. We evaluate the performance of LMs with two different sizes: 1) RoBERTa-large (with 340 million parameters) and 2) RoBERTa (with 110 million parameters). The results show that using a larger LM leads to significant improvement in performance, with an increase of 11.71% and 14.30% in model accuracy on the IHdev dataset and the IHtest dataset, respectively.

Table 6 shows the impact of the knowledge component of `LMExplainer`. We compare the performance of the LM-only model with and without external knowledge from ConceptNet. *only* means we only use the LM to predict the answer. *+ external knowledge* means the external knowledge is leveraged. We observe that incorporating external knowledge significantly improves the accuracy of the LM prediction, especially on the test set. With external knowledge, the model accuracy on IHdev and IHtest is increased by at least 3.69% and 7.12%, respectively.

In Table 4, we analyze the impact of the interpreting component on LM performance. *w/o itp* indicates that the interpreting component was not incorporated in the prediction, whereas the *+ itp* indicates its presence. We observe that removing the interpreting component leads to a clear decrease in accuracy by at least 4.92% and 5.35% on IHdev and IHtest, respectively. Furthermore, comparing the results of *RoBERTa-large only*, *RoBERTa-large + itp*, and *final*, we find that the interpreting component has a greater impact on accuracy than the other components.

The ablation highlights the positive contributions of each component of `LMExplainer`. Specifically, we find that the interpreting component plays a crucial role in enhancing model accuracy and generalizability on unseen questions.

## 5 LIMITATION

In the pursuit of transparency and rigorous discourse, we recognize several limitations in our method. Our KG relies on ConceptNet. Any limitations or inaccuracies within ConceptNet could impact the quality and accuracy of our explanations. Another potential constraint is that the explanations generated by our selected LM could vary if other LMs were used, potentially affecting the content and style of the explanations.

## 6 CONCLUSIONS

In this paper, we propose `LMExplainer`, a novel model that incorporates an interpretation module to enhance the performance of LMs while also providing clear and trustworthy explanations of the model's reasoning. Our explanation results are presented in a logical and comprehensive manner, making it easier for humans to understand the model's reasoning in natural language. Our experimental results demonstrate superior performance compared to prior SOTA works across standard datasets in the commonsense domain. Our analysis shows that `LMExplainer` not only improves the model's performance but also provides humans with a better understanding of the model.

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

## A.1 OTHER EXPLANATION EXAMPLES

We demonstrate the complete explanation example of PathReasoner and ECQA in Table 7. These methods exhibit in an unclear and intricate manner. Such explanations make it hard for humans to understand the decision-making process behind the model.

| Input Questions | Q: What is someone doing if he or she is sitting quietly and his or her eyes are moving? A. reading B. meditate C. fall asleep D. bunk E. think |
|---|---|
| **Label** | A. reading |
| | **Explanation of Others** |
| **Path-Reasoner** | quietly [related to] quiet [at location] a library [used for] reading eyes [used for] reading eyes [form of] eye [related to] glasses [used for] reading sitting [related to] sit [related to] relaxing [has subevent] reading |
| **ECQA** | **Positive examples:** - When we read, our eyes move. - While reading, a person sits quietly, **Negative examples:** - While meditating, eyes don't move, eyes are closed, - While sleeping, eyes are closed and they don't move, - When a person bunks, he/she doesn't sit quietly, - Eyes don't move when you think about something. **Explanation:** When we read, our eyes move. While reading, a person sits quietly. While meditating and sleeping, eyes don't move, eyes are closed. When a person bunks, he/she doesn't sit quietly. Eyes don't move when you think about something. |

Table 7: The complete explanation examples of PathReasoner and ECQA.

## B.2 DETAILS OF ELEMENT-GRAPH

Due to space constraints in the main text, we provide a comprehensive description of the node and relations types, alongside the detailed equations for computing their embeddings.

The node-type $u_i$ are the one-hot vectors of the node types. The type is according to the node's origin form, the input content $z$, question $\{q\}$, answer $\mathcal{A}$, or the node in the KG. The $u_i$ is transformed into an embedding through a linear transformation for subsequent calculations.

The relation type $r_{ij}$ is determined using pre-defined templates, which are employed to extract relations from the knowledge triples in the KG (Feng et al., 2020). The embedding $\boldsymbol{r}_{ij}$ for the relation is computed for subsequent use:

$$\boldsymbol{r}_{ij} = f_\zeta(r_{ij}, u_{ij}) = f_\zeta(r_{ij}, u_i, u_j), \tag{6}$$

where $f_\zeta$ is a two-layer MLP, $u_{ij}$ denotes the concatenation of $u_i$ and $u_j$.

The node score $v_{score}$ is subsequently utilized in its embedded form, calculated by:

$$\boldsymbol{v}_{score} = f_\rho(v_{score}) \tag{7}$$

where $f_\rho$ is a two-layer MLP.

## C.3 IMPLEMENTATION DETAILS

Due to the constraints of space, we present the specific details of our explanation generator here.

For explanation generation, the example instruction we used in the first stage is *Q="Question content is"*, *A="The predicted choice is"*, *R="According to the model top reason-elements"* + $\mathcal{K}$ + *"explain the model reasoning process with 'since. . .' "*, $\mathcal{K}$ is the top reason-elements of the model. In the second stage, *P="According to"*, and *T="explain why the model doesn't choose other options with 'The other potential choices' "*.

## D.4 CASE STUDY

In this section, we will present more explanation examples generated by LMExplainer, to demonstrate the effectiveness of our method. These questions are randomly selected from the dataset. By sharing these examples, we aim to highlight the broad applicability and robust performance of LMExplainer across diverse contexts. These examples not only illustrate the adaptability of our approach but also substantiate its validity by demonstrating consistent results.

**[Example A]**

**Input Question:** Where could there by tens of thousands of people playing baseball?

A. America B. Canada C. Pitcher's mitt D. Park E. Sporting goods store

**Label:** A. America

**Ranked Reason-elements:** 1. bronx new york, 2. amharic, 3. baseball game, 4. pitcher's mitt, 5. thousand and one.

**Why-choose explanation:**

Since the reason-elements bronx new york, baseball game, and thousand and one are associated with American baseball culture and history, and pitcher's mitt is specifically related to playing baseball, the model predicted that the answer is America where there could be tens of thousands of people playing baseball.

**Why-not-choose explanation:**

The other potential choices , Canada, park, and sporting goods store , are unlikely because these locations do not have the same historical significance or cultural association with baseball as America. Additionally , the word pitcher's mitt suggests an outdoor area where people can play baseball, making it even less likely that a sporting goods store would be the correct answer.

**[Example B]**

**Input Question:** What would you see if there were different types of flowers near you?

A. Many colors B. Smell good C. Cover fields D. Smell bad E. Continue to grow

**Label:** A. Many colors

**Ranked Reason-elements:** 1. plant themselves, 2. flower arrangement, 3. pair of shoes, 4. female reproductive part of flower, 5. wisconsin product.

**Why-choose explanation:**

Since the model identified flower arrangement and female reproductive part of the flower as

top reasons, it suggests that the presence of different types of flowers would likely result in a variety of colors and arrangements of petals, which aligns with the predicted option of many colors.

**Why-not-choose explanation:**

The other potential choices , such as smell good and smell bad , are excluded because the model did not identify any relevant keywords related to scent. Similarly , cover fields and continue to grow are not directly related to the presence of different types of flowers, which is the focus of the question.

**[Example C]**

**Input Question:** Why does someone engage in chatting with friends when they want to workout?

A. Fun B. Agreement C. Cell phone D. Answers E. Inspiration

**Label:** E. Inspiration

**Ranked Reason-elements:** 1. assemble cleaning materials, 2. gain, 3. feeling of comfort, 4. chatoyance, 5. walking.

**Why-choose explanation:**

Since the words gain and feeling of comfort suggest a desire to improve oneself during a workout, and assemble cleaning materials and walking imply a pre-workout routine, the model predicts that the reason for engaging in chatting with friends during a workout is inspiration, as it can provide motivation and support to achieve fitness goals.

**Why-not-choose explanation:**

The other potential choices , such as fun, agree- ment , and cell phone , are unlikely reasons for engaging in chatting during a workout because they do not offer a clear connection to exercise. Additionally , they do not address the underlying motivation for the workout or the desire to improve oneself. Similarly , the word answers does not align with the context of working out and could be interpreted in various ways, making it an improbable option.

