# OpenReview forum: "LMExplainer: A Knowledge-Enhanced Explainer for Language Models"
_ICLR.cc/2024/Conference — ICLR 2024 Conference Withdrawn Submission_

### Official Review · Reviewer_hNAs · 2023-10-25

**Soundness:** 2 fair
**Presentation:** 2 fair
**Contribution:** 2 fair
**Rating:** 3
**Confidence:** 3

**Summary:**

This paper studies the interpretability issue of large language models.
Different from some existing studies that use attention-based methods, the authors introduce knowledge graphs to generate explanations.
Specifically, the key decision signals (elements in the constructed knowledge graph) are extracted and used as instructions for large language models to generate explanations.
The experimental results reflect the explanations could help model accuracy.

**Strengths:**

- The interpretability of LLMs deserves to be investigated and the proposed approach is model-agnostic.
- Knowledge graphs are leveraged to extract key elements to support the LLMs to generate explanations.
- The experimental results show both good model accuracy and interpretability.

**Weaknesses:**

- Although the proposed approach is model-agnostic, it is only tested on RoBERTa-large and its variants. Larger language models are suggested to be tested. This is because LLMs have strong capacity and knowledge graphs might not help them.
- The approach is not clearly explained. How to optimize f_{enc} and MLP is not stated. Moreover, how the explanations affect the model accuracy is not clear. As shown in Figure 1, the generated explanations do not affect answer selection. Besides, how to optimize the whole method?
- The baselines seem to be not very new. More recent and strong baselines are preferred.
- From my personal perspective, the novelty of the techniques are not well validated.

Typos:
(1) "their decision process lack transparency" -> lacks

**Questions:**

Please see the weakness part.

---

> ### Author Response · Authors · 2023-11-20
>
> We sincerely appreciate your thorough review and insightful feedback on our manuscript. Your comments have not only helped us in refining our work but also provided an opportunity to clarify and elaborate on key aspects of our research. Below, we address each of your points in detail.
>
>
> **Response to Weaknesses:**
>
> **1. (W1) Testing on Larger LM:**
>
> In our current experiments, we selected RoBERTa-large and its variants as our primary test subjects due to their effectiveness and widespread recognition in the NLP community. We acknowledge the impressive capabilities of larger LMs like GPT-3.5 in language understanding and generation. However, we posit that integrating KGs with these models can yield substantial benefits. KGs provide structured, domain-specific knowledge that may not be entirely encompassed by the pre-trained parameters of LLMs. This integration is particularly advantageous in tasks requiring specific knowledge domains.
>
> To further support this claim, we conducted a zero-shot experiment with GPT-3.5-turbo, maintaining the same experimental settings as our tests on the CommonsenseQA dataset. The results were telling: GPT-3.5-turbo achieved an accuracy of 73.54%, while our approach yielded a higher accuracy of 77.31% (Table below).
> | Model            | Accuracy (%) |
> |------------------|--------------|
> | GPT-3.5-turbo    | 73.54        |
> | **LMExplainer (ours)**| 77.31        |
>
> This outcome indicates that the performance of advanced LLMs like GPT-3.5 may not be as robust as anticipated. This underscores the necessity and relevance of our approach that integrates KGs, demonstrating its potential to fill gaps where even sophisticated LLMs might fall short.
>
> We want to emphasize the primary benefit of our graph modular design is the transformation of the 'black-box' nature of LM into a more transparent graph structure. By narrowing down the knowledge search space of the LM, our approach provides a clearer understanding of the model's decision-making process and contributes to improve model performance. Our approach demonstrates that it is possible to achieve both high performance and high explainability in LMs, addressing a key challenge in the field of AI.
>
>
> **2. (W2) Clarification of Approach:**
>
> The MLP in Section 3.2 is a learned part of the pre-trained LM. This design employs the probability computation function of the pre-trained LM to calculate the score. To reduce the misunderstanding, we have revised equation (2), and added a clear description in the updated version. Please review the latest version. The recently modified sections are highlighted in blue for easy identification.
>
> The MLPs in the later section are trained with the LM. When LM is fine-tuned, the MLPs are trained simultaneously. The loss function used is cross-entropy loss. The dimension of hidden layers is 200, and the hidden layers are 2. The optimization involves fine-tuning both the knowledge representation (via the GAT) and the predictive components (encoder and MLPs).
>
> In our design, explanations are generated based on the predicted answer and key reason-elements. The explanations are not used to affect the answer selection. The explanations are generated to reflect the decision-making process of the model in a human-understandable way.
>
> **3. (W3) Clarification of Baselines:**
>
> We have included what we believe are the latest and most typical works as baselines. This selection was made to provide a comprehensive benchmark. We recognize that the field is continuously evolving, and we are dedicated to adapting our work to reflect these changes. Currently, we are committed to conducting a thorough review and update of our baseline. We will ensure that our study remains relevant and provides a meaningful contribution to the field. If there are specific recent works or baselines that you believe would enhance the robustness of our comparison, we kindly invite you to list them.
>
>
> **It's not over yet, please check our next comment.**

---

> > ### Author Response · Authors · 2023-11-20
> >
> > **4. (W4) Validation of Novelty:**
> >
> > We understand the importance of clearly demonstrating the innovative aspects of our work and its contribution to the field. To address your concern, we would like to highlight the following points:
> > - **Transformation to a Transparent Structure:** A central innovation of our approach is the transformation of the traditionally 'black-box' nature of LMs into a more transparent and interpretable structure. This transformation is crucial for understanding the decision-making process of LMs, marking a significant departure from conventional methods.
> > - **Bridging Model Reasoning with Human-Understandable Language:** Our method bridges the gap between complex model reasoning and human comprehension. By translating the model's reasoning into natural, human-understandable language, our approach not only enhances interpretability but also paves the way for future LMs to build greater trust between models and users. This aspect of our work is particularly important in making AI more accessible and trustworthy.
> > - **Human-Centric Explanations:** Distinguishing our approach from others, we focus on human-centric explanations. Our method prioritizes ease of understanding and comprehensiveness, enabling users to grasp the model's reasoning intuitively. This human-centric approach is a significant stride towards making AI explanations more user-friendly and practical.
> > - **Adaptability to Other LMs:** Another novel aspect of our work is the adaptability of our interpreting design. It can be easily integrated with other LMs, showing the flexibility and applicability of our approach across various LMs.
> >
> > ---
> >
> > We are grateful for the opportunity to enhance our manuscript based on your feedback. We hope that our response and revised version will address your concerns and improve the quality of our work. We look forward to hearing from you.

---

> ### Author Response · Authors · 2023-11-22
> **Look Forward to Your Response**
>
> Dear Reviewer,
>
> Since we are on the last day of author-reviewer discussions, we keenly await your feedback on our rebuttal and the paper modifications to address your comments. We believe we have addressed all the concerns raised by you. If there are any outstanding concerns, please let us know.
> We look forward to your response and appreciate any feedback.
>
> \
> Thank you,
>
> Authors

---

### Official Review · Reviewer_d4rH · 2023-10-29

**Soundness:** 3 good
**Presentation:** 3 good
**Contribution:** 3 good
**Rating:** 6
**Confidence:** 3

**Summary:**

Language models like GPT-4 are powerful for natural language processing tasks but lack transparency due to their complex structures, which can hinder user trust. To address this issue, LMExplainer is proposed, which is a knowledge-enhanced explainer that provides human-understandable explanations by locating relevant knowledge in a large-scale knowledge graph using graph attention neural networks. Experiments show that LMExplainer outperforms existing methods on CommonsenseQA and OpenBookQA datasets, generating more comprehensive and clearer explanations compared to other algorithms and human-annotated explanations.

**Strengths:**

1. The proposed model is clear and easy to follow.
2. The studied topic of explaining LLM with knowledge graph is a reasonable direction. The idea of improve accuracy and explanation simutaneously is interesting.
3. The experimental results are sound.

**Weaknesses:**

1. The code is not available to the public.
2. The modular design of proposed model made it hard to judge where the benefits of the model is from.

**Questions:**

1. It seems the proposed model is not learned in an end-to-end way. How was the parameters of MLP learned in line 5 of Algorithm 1? and how does the generated graph influence the final results?

---

> ### Author Response · Authors · 2023-11-20
>
> We sincerely appreciate your thorough review and insightful feedback on our manuscript. Your comments have not only helped us in refining our work but also provided an opportunity to clarify and elaborate on key aspects of our research. Below, we address each of your points in detail.
>
>
> **Response to Weaknesses:**
>
> **1. (W1) Availability of Code:**
>
> We care about the reproducibility of our work very much. It is our responsibility to provide the code and data to the community. We are currently preparing our code for public release. We will ensure that it is well-documented and accessible to the community.
>
> **2. (W2) Modular Design:**
>
> Thank you for your comment regarding the modular design of our proposed model. We appreciate the opportunity to clarify how this design contributes to the model's effectiveness and transparency.
>
> The primary benefit of our model's modular design is the transformation of the 'black-box' nature of LM into a more transparent graph structure. By narrowing down the knowledge search space of the LM, our approach provides a clearer understanding of the model's decision-making process and contributes to improve model performance. This transparency is crucial for applications where understanding the rationale behind model predictions is as important as the predictions themselves.
>
> We include an ablation study analyzing the main components of our model to further validate the benefits of our design in the paper. The results from this study clearly indicate that our interpretation design, a core aspect of the modular approach, is instrumental in enhancing the final performance of the model. This study provides empirical evidence supporting the advantage of our design.
>
>
> **Response to Questions:**
>
> **1. (Q1) Design and Influence Clarification:**
>
> Our model operates in an end-to-end manner in terms of obtaining key elements from the reasoning process for explanation purposes. Our model extracts these elements and integrates them into a constrained structure, which is then processed by the GPT-3.5-turbo. The GPT-3.5-turbo is utilized primarily for organizing these elements into a human-understandable way.
>
> The MLP in Algorithm 1 is a learned part of the pre-trained LM. This design employs the probability computation function of the pre-trained LM to calculate the score. To reduce the misunderstanding, we have revised equation (2), and added a clear description in the updated version. Please review the latest version. The recently modified sections are highlighted in blue for easy identification.
>
> The generated graph provides the structured representations for knowledge. Such structure is more interpretable. The graph is also used to guide the reasoning process of the LM. The LM is constrained to search for knowledge from the graph, which narrows down the knowledge search space and improves the performance. The accuracy and relevance of the retrieved graph are important, as they directly impact the LM's performance in generating predictions or explanations. By ensuring that the graph accurately captures the relative knowledge, our model effectively utilizes these insights for decision-making and explanation generation.
>
> ---
> We are grateful for the opportunity to enhance our manuscript based on your feedback. We hope that our response and revised version will address your concerns and improve the quality of our work. We look forward to hearing from you.

---

> ### Author Response · Authors · 2023-11-22
> **Look Forward to Your Response**
>
> Dear Reviewer,
>
> Since we are on the last day of author-reviewer discussions, we keenly await your feedback on our rebuttal and the paper modifications to address your comments. We believe we have addressed all the concerns raised by you. If there are any outstanding concerns, please let us know.
> We look forward to your response and appreciate any feedback.
>
> \
> Thank you,
>
> Authors

---

### Official Review · Reviewer_gb89 · 2023-10-30

**Soundness:** 2 fair
**Presentation:** 2 fair
**Contribution:** 2 fair
**Rating:** 5
**Confidence:** 3

**Summary:**

Language models like GPT-4 are powerful but opaque. To make their decisions understandable, a tool called LMExplainer has been developed. It harnesses Knowledge Graphs and the Graph Attention Neural Network (GAT) to pinpoint and explain the model's reasoning. In tests on two QA datasets, LMExplainer outperformed benchmarks and adeptly translated the model's logic into plain language.

**Strengths:**

- The authors adeptly use natural language to elucidate the inference results of LMs.
- LMExplainer was tested on QA datasets and benchmarked against leading models. The results indicated a superior performance of LMExplainer in most scenarios.

**Weaknesses:**

- The notation could be improved for clarity, especially inconsistencies like h_e^k and h_{v_e}^k in Equation (5) and Equation (6).
- Equations in section 3 lack immediate clarity. Incorporating variable and function dimensions would enhance readability.
- The study exclusively utilizes RoBERTa for LMExplainer. It's uncertain if similar results would be achieved with different LMs.
- The structure of the experiment section needs refinement. In tables like Table 1 and Table 2, clearer categorization of the baseline models, such as which belong to fine-tuned LM RoBERTa-large or KG-augmented RoBERTa-large, would be beneficial.

**Questions:**

- In Equation 5, the authors use f to represent MLP as m_{es}=f_n(,,). What does the “comma” signify? Is it indicating concat, inner product, element-wise addition, or another operation?
- What does \hat{n} in Equation 5 represent? The authors haven't previously defined it.
- In Algorithm 1, line 4, the authors apply f_{enc} to encode node v and use the encoded embeddings for KG extraction. Is f_{enc} equivalent to the LM - Encoder in Figure 1? If yes, it would be clearer if there were an arrow connecting LM - Encoder and KG Retrieval in the figure.
- The term u^e is derived from a linear transformation of one-hot node-type vectors. What are these node-type vectors? An example would help in understanding.

---

> ### Author Response · Authors · 2023-11-20
>
> We sincerely appreciate your thorough review and insightful feedback on our manuscript. Your comments have not only helped us in refining our work but also provided an opportunity to clarify and elaborate on key aspects of our research. Below, we address each of your points in detail.
>
>
> **Response to Weaknesses:**
>
> **1. (W1) Notation Clarity:**
>
> Thank you for pointing out the inconsistency in our notation between Equations (5) and (6). In the mentioned equations, the notations h^k_e and h^k_{v_e} were intended to represent the same concept. We apologize for the confusion caused by this typo.
>
> We noticed that Section 3.3 was potentially confusing to readers. In response, we have actively adopted your suggestions and made corresponding revisions. We have optimized the paragraph structure and improved the presentation of the notation to ensure the content is concise and clear. Regarding the content previously found in equations (5) and (6), we have now moved it to Appendix B.2 and provided a more detailed explanation. Please review the latest version. The recently modified sections are highlighted in blue for easy identification.
>
> We have ensured that the notation accurately reflects the intended meaning and enhances the overall clarity of the mathematical expressions.
>
> **2. (W2) Clarity of Equations in Section 3:**
>
> We understand that the readability and comprehensibility of mathematical expressions are important for the effective communication of our methodology. To address your suggestion, we have specified the dimensions of each variable and the input-output dimensions of each function in the revised version. Please review the latest version. The modified sections are highlighted in blue.
>
>
> **3. (W3) Utilization of LM:**
>
> While our current study focuses on RoBERTa, we designed LMExplainer with the intention of it being a universal method, applicable to a wide range of LMs. The core mechanisms of LMExplainer, such as the integration of knowledge graphs and the interpreting component, are not specific to RoBERTa. These components are designed to be adaptable to the architectures and functionalities of various LMs. Moreover, the method of extracting and utilizing knowledge from KGs, and the way explanations are generated and validated, are largely model-agnostic and can be applied to other LMs with minimal modifications. We acknowledge that empirical validation with other models is necessary to fully demonstrate the universality of LMExplainer. As part of our future work, we plan to extend our experiments to include a variety of LMs, to validate the adaptability of our method.
>
> **4. (W4) Table Structure:**
>
> All models presented in Tables 1 and 2 are integrations of LM and KG. In our revised version, we will add detailed legends and footnotes to these tables to provide additional context and clarification.
>
> **Response to Questions:**
>
> **1. (Q1) Comma Notation:**
>
> In previous Equation 5, the commas within the function f_n(,,) denote the concatenation of inputs. We have clarified this in our latest version, and ensure that the notation throughout the manuscript is consistent and correct.
>
> **2. (Q2) Clarification of \hat{n}:**
>
> In the previous version, \hat{n} means the embedding of node scores. We have actively adopted your suggestions and made corresponding revisions. We have reduced the overload of symbols to lessen reader strain, simplified our descriptions, and moved the detailed implementation content to the appendix. Please refer to Section 3.3 and Appendix B.3 to review our modification.
>
> **3. (Q3) Function f_{enc}:**
>
> Yes, f_{enc} is equivalent to the LM-Encoder in Figure 1. We will add an arrow connecting LM-Encoder and KG Retrieval in the figure.
>
> **4. (Q4) Node-type Vectors:**
>
> The node-type vectors are the one-hot vectors of the node types. The type is according to the node's origin form, the input content $z$, question $\{q\}$, answer $\mathcal{A}$, or the node in the KG. We have added the details of node-type information in the latest version (Appendix B.2).
>
> ---
>
> **Our main responses to your feedback are incorporated in the latest version. We kindly hope you review these updates. All modifications have been highlighted in blue for easy identification.**
>
> We are grateful for the opportunity to enhance our manuscript based on your feedback. We hope that our response and revised version will address your concerns and improve the quality of our work. We look forward to hearing from you.

---

> ### Author Response · Authors · 2023-11-22
> **Look Forward to Your Response**
>
> Dear Reviewer,
>
> Since we are on the last day of author-reviewer discussions, we keenly await your feedback on our rebuttal and the paper modifications to address your comments. We believe we have addressed all the concerns raised by you. If there are any outstanding concerns, please let us know.
> We look forward to your response and appreciate any feedback.
>
> \
> Thank you,
>
> Authors

---

### Official Review · Reviewer_gatd · 2023-10-30

**Soundness:** 2 fair
**Presentation:** 3 good
**Contribution:** 2 fair
**Rating:** 5
**Confidence:** 4

**Summary:**

This paper introduces a knowledge-enhanced explainer, LMExplainer. First, LMExplainer constructs a subgraph by integrating external knowledge into the tokens of the question. Subsequently, it utilizes graph attention neural networks to represent the subgraph and identify the most critical reasoning components within it. Finally, LMExplainer leverages a predefined structure containing the extracted explanation components. These components are used to prompt GPT-3.5 to generate explanations regarding why LMs selected a specific answer and why they rejected others. Experimental results from CommonsenseQA and OpenBookQA datasets demonstrate the effectiveness of the proposed method.

**Strengths:**

1. The idea of utilizing a knowledge graph as a more transparent surrogate to generate comprehensible explanations seems reasonable. This can, to some extent, reveal the reasoning process of language models.
2. The proposed approach that extracts reason-elements as one part of the key components which are essential to the LM decision-making process is somewhat novel.
3. The experiments and subsequent analyses demonstrate that LMExplainer not only enhances the performance of language models in open domain question answering tasks, but also exhibits the capacity to generate comprehensible explanations for these models.

**Weaknesses:**

1. In Section 3.2, the paper employs all the tokens in the question sentence as queries to retrieve relevant knowledge. However, if there are non-entity tokens such as "what," "is," and so on, they can introduce noise into the process of constructing subgraphs. This could potentially increase the time and space complexity of the graph construction algorithm. As far as I am concerned, it is imperative to filter out non-entity words from the question prior to constructing subgraphs.
2. In constructing subgraphs, the authors extract L-hop neighbors of z, indicating that 'L' is a crucial factor that influences the introduction of an appropriate number of knowledge. However, the authors did not specify how 'L' was set and did not conduct ablation experiments to investigate how 'L' affects model performance.
3. In the ablation studies, the experimental results reveal a significant decrease in accuracy when the interpreting component in LMExplainer is removed. However, in Section 3.4, the authors note that the explanations are generated using the predicted answer. Consequently, it appears that the accuracy of model predictions is not influenced by whether or not explanations are generated. The authors should therefore clarify the relationship between generating explanations and answer predictions.
4. There are no human evaluations of the accuracy of the explanations generated by GPT-3.5, which weakens the case presented in Table 3. This could potentially be a rare occurrence where the generated explanations happen to be correct. Further studies should be conducted to ascertain the proportion of accurately generated explanations.

**Questions:**

1.  In Section 3.2, the authors explain that the score of each node, which represents the correlation between node v and input content, is derived by passing token embeddings through an MLP. Could you please provide additional information on how the MLP is trained?
2. There are numerous traditional methods that also integrate knowledge graphs into language models. Undoubtedly, incorporating an external knowledge module can greatly enhance the performance of the model. Could you please elaborate further on the advantages of your knowledge integration method?
3. This paper employs predefined structures to prompt GPT-3.5 to produce explanations. However, given the propensity of large models to suffer from hallucination, it is highly likely that they will generate explanations that are unrelated to the question, despite the presence of relevant words in the prompt. This appears to contradict the statement made in the Conclusion section, which asserts that the model can generate trustworthy explanations. How can you ensure that the generated explanations are indeed trustworthy?

---

> ### Author Response · Authors · 2023-11-20
>
> We sincerely appreciate your thorough review and insightful feedback on our manuscript. Your comments have not only helped us in refining our work but also provided an opportunity to clarify and elaborate on key aspects of our research. Below, we address each of your points in detail.
>
>
> **Response to Weaknesses:**
>
> **1. (W1) Token Filtering in Subgraph Construction:**
>
> We would like to clarify that our methodology, in fact, aligns with the approach recommended by the reviewer.
>
> In our implementation, we have adopted a token filtering process following Yasunaga et al (2021). This approach specifically involves the exclusion of non-entity words, such as common stopwords ("what," "is," etc.), from the queries used to retrieve relevant knowledge. By doing so, we effectively mitigate the risk of introducing noise into the subgraph construction and maintain the efficiency of our algorithm in terms of both time and space complexity.
>
> **2. (W2) Clarification on 'L':**
>
> In our current implementation, the choice of 'L' was based on preliminary experiments and domain expertise. Our setting aims to balance the depth of the captured knowledge against the risk of overloading the model with irrelevant information. Specifically, a larger 'L' introduces many noise nodes, decreasing the quality of the extracted knowledge. On the other hand, a smaller 'L' may not capture sufficient knowledge to support the model's reasoning. In our experiments, we found when 'L=3', an average of over 400 nodes were included in the subgraph. This number is sufficient to capture the relevant knowledge for building element-graph, and effectively narrow down the knowledge search space.
>
> We plan to further elaborate on this in the future revised version.
>
>
>
> **3. (W3) Significance of the Interpreting Component:**
>
> The interpreting component is designed to reflect the reasoning process of the LM. It is the model's internal representation (from GAT) that contributes to the final prediction.
> This component is important in transforming the reasoning of the model into an interpretable and human-understandable format. Our approach to generating explanations is deeply rooted in the interpreting component. It uses the model's predicted answer as a basis and leverages the interpreting component to trace back the reasoning process.
>
> **1. (W4) Human Evaluations of Explanation Accuracy:**
>
> We understand the significance of such evaluations in validating the accuracy of our explanations. To address this, we have conducted a human study using a crowdsourcing platform. We will discuss this part in our revised version.
> Below are the key details and findings of our study:
> - **Participants Statistics:** Our human evaluations were conducted on the Prolific (https://www.prolific.co) platform. We had 50 participants who were native English speakers, with a minimum education level of high school. The participant group was balanced in terms of gender, and 54% of them held an undergraduate degree or higher. This diverse participant pool ensures that our findings are broadly representative.
> - **Evaluation Process and Metrics:** Participants were provided with detailed instructions and examples to maintain consistent rating standards. They evaluated 20 randomly selected QA from our test dataset. Each QA has our generated explanations. Our evaluation metrics included overall quality, understandability, trustworthiness, detail sufficiency, completeness, and accuracy. Each explanation was accompanied by evaluation questions, with responses gathered using a three-point (1-3) Likert scale (higher is better).
> - **Results:** The results of our human evaluation are as follows:
>
>     | Evaluation Metric      | Score    |
>     |------------------------|----------|
>     | Overall Quality        | 0.845    |
>     | Understandability      | 0.89     |
>     | Trustworthiness        | 0.856    |
>     | Sufficiency of Detail  | 0.831    |
>     | Completeness           | 0.811    |
>     | Accuracy               | 0.845    |
>
>     These scores was normalized to the range [0, 1], indicating a high level of effectiveness and accuracy in the explanations generated by our approach.
>
> - **Insights:** The majority of our participants (90%) have used AI, and 92% are under the age of 35. This demographics aligns well with real-world using scenarios, suggesting that our human-centric explanations have significant potential for future human-centered applications.
> - **Efforts to Minimize Bias:** While completely eliminating bias is challenging, we made concerted efforts to minimize it. We followed the methodology outlined in Hoffman et al. (2018) [1] for cross-human alignment and evaluation questions design.
>
> ---
> [1] Hoffman, Robert R., et al. "Metrics for explainable AI: Challenges and prospects." arXiv preprint arXiv:1812.04608 (2018).
>
>
> **It's not over yet, please check our next comment.**

---

> ### Author Response · Authors · 2023-11-20
>
> **Response to Questions:**
>
> **1. (Q1) Equation in Section 3.2:**
>
> The MLP in Section 3.2 is a learned part of the pre-trained LM. This design employs the probability computation function of the pre-trained LM to calculate the score. To reduce the misunderstanding, we have revised equation (2), and added a clear description in the updated version.
> Please review the latest version. The recently modified sections are highlighted in blue for easy identification.
>
> **2. (Q2) Our Advantage of Knowledge Integration:**
>
> 1. LMExplainer addresses the challenge of integrating large-scale KGs with LMs. We utilize GATs to efficiently obtain the key reasoning elements. The GATs are trained with the LM, which learns the reasoning patterns of the LM.
>
> 2. Central to our approach is a transparent surrogate model, deeply rooted in the KG. This model reflects the reasoning process of LMs, offering a clear window into their decision-making mechanisms and increasing interpretability.
>
> 3. Furthermore, the integrated knowledge from KGs constrains and guides the reasoning process. This enables us to map the complex reasoning of LMs onto a graph structure, which we then translate into explanations that are easily understandable by humans. Our method transforms intricate decision-making processes into clear, comprehensible explanations, making the inner reasoning of LMs more accessible than ever before.
>
> **3. (Q3) Addressing Hallucination Issues:**
>
> Hallucination is a common problem and big topic in LLMs. We are not aiming to remove all the hallucinations but using strong constraints to narrow down the knowledge search space of LLMs, to maintain the trustworthy. The use of predefined structures in prompting is designed to guide the model towards generating more relevant and focused explanations, thereby reducing the hallucinated content.
>
> We have also conducted a human evaluation to verify the quality of the explanations (see W4). The results confirm the high quality and reliability of the explanations generated by LMExplainer. Example feedback included comments such as: "In comparison to prior explanations, these explanations provide a more intuitive understanding of the model's decision-making process. The explanations are cogent, and even in instances of erroneous predictions, the underlying reasoning remains transparent and comprehensible."
>
> ---
> We are grateful for the opportunity to enhance our manuscript based on your feedback. We hope that our response and revised version will address your concerns and improve the quality of our work. We look forward to hearing from you.

---

> ### Author Response · Authors · 2023-11-22
> **Look Forward to Your Response**
>
> Dear Reviewer,
>
> Since we are on the last day of author-reviewer discussions, we keenly await your feedback on our rebuttal and the paper modifications to address your comments. We believe we have addressed all the concerns raised by you. If there are any outstanding concerns, please let us know.
> We look forward to your response and appreciate any feedback.
>
> \
> Thank you,
>
> Authors

---

> > ### Comment · Reviewer_gatd · 2023-11-22
> > **Thanks for the authors responses**
> >
> > Dear Authors,
> >
> > Thank you for your detailed response to our comments and your efforts in addressing the concerns identified during the initial review. It is acknowledged that your responses have resolved some of my initial queries. The additional experimental work conducted to validate the accuracy of the explanations has strengthened the paper to some extent.
> >
> > However, the question of how the interpreting component has an impact on the accuracy of the prediction has not been addressed carefully yet: After the model predicts answers, the interpreting component uses them to generate explanations, but how do the generated explanations influence the accuracy? Besides, though the authors provide some discussions on the advantages of the knowledge integration method, I still don't feel that the proposed GAT-based training approach is novel enough.
> >
> > In summary, I will keep my evaluation.

---

> ### Author Response · Authors · 2023-11-22
>
> Dear Reviewer,
>
> Thank you for acknowledging our efforts in addressing the initial concerns and for your continued engagement with our work. We appreciate your feedback and would like to provide additional clarification on the points you raised.
>
> The interpreting component in our model plays an important role in enhancing the transparency and understandability of the predictions made by the LMs. We demonstrated and analyzed how the interpreting component influences the performance in our ablation study. Please note that our generated explanations will NOT influence the performance of the LMs. The explanation is used to help human understand the reasoning process of the LMs. The LMs will not be trained on the explanations.
>
> We understand your concerns regarding the novelty of our GAT-based training approach. To further emphasize our noval aspects, we would like to highlight the following points:
> 1. We utilize GATs to provide a more transparent and interpretable decision-making process. Our main contribution is not proposing a new GAT architecture to help LM training. Instead, we leverage the KG and build a GAT-based surrogate to demystify the reasoning process of LMs. However, when we investigate the performance with our designed GAT fine-tuning approach, we found that it can also improve the performance of the LM. This is an interesting finding and we believe that it is worth mentioning in our paper.
>
> 2. We want to emphasize that our novelty lies in **(1) explaining how LMs work by generating explanations on a more transparent surrogate, and (2) providing human-understandable and faithful explanations for LMs' decision-making process.**
> Here is an example in XAI:
>
>     ``
>     Q: What is someone doing if he or she is sitting quietly and his or her eyes are moving?
>     ``
>
>     ``
>     A. reading B. meditate C. fall asleep D. bunk E. think
>     ``
>
>     ```
>     Explanation of [Path-Reasoner [1]]:
>
>     quietly [related to] quiet [at location] a library [used for] reading eyes [used for] reading,
>     eyes [form of] eye [related to] glasses [used for] reading,
>     sitting [related to] sit [related to] relaxing [has subevent] reading
>     ```
>     Our results:
>     ```
>     Explanation of [LMExplainer] (ours):
>
>     Ranked Reason-elements:
>     1. quiet chattering mind, 2. not making sound, 3. mind focuses, 4. glasses for people with poor eyesight, 5. war
>
>     Explanation (why-choose):
>     Since the person is described as sitting quietly and their eyes are moving, it is likely that they are engaged in a visual activity.
>     Based on the keyword “glasses for people with poor eyesight”, option “A. reading” is the most likely answer, as reading is a common visual activity that requires focusing one’s eyes on a page and is often aided by glasses for people with poor eyesight.
>
>     Explanation (why-not-choose):
>     The other options , such as “B. meditate” or “C. fall asleep”, involve closing one’s eyes or having a still mind,
>    so it is unlikely that the person is doing either of those activities if their eyes are moving.
>     Similarly, “D. bunk” and “E. think” do not seem to be related to the visual activity of
>     having one’s eyes move while sitting quietly.
>     ```
>
>     Although Path-Reasoner is a very advanced approach in this area, it is still difficult for human to understand the reasoning process of the LM. And most works in this area did the same things.
>
>
>     To the best of our knowledge, LMExplainer first work capable of leveraging graph-based knowledge in generating natural language explanations on the rationale behind LM behaviors.
>
> We hope these additional explanations address your concerns. We are open to any further suggestions or queries you may have.
>
> \
> Thank you,
>
> Authors
>
> ---
> [1] Xunlin Zhan, Yinya Huang, Xiao Dong, Qingxing Cao, and Xiaodan Liang. Pathreasoner: Explainable reasoning paths for commonsense question answering. Knowledge-Based Systems, 235: 107612, 2022a.

---

### Author Response · Authors · 2023-11-15
**Official Comment**

We sincerely apologize for the accidental upload of deleted comments. We are dedicatedly working on refining the rebuttal and ensuring that it addresses the reviewers' questions and concerns. We kindly request you to refer to the latest version of our official comments, which more accurately reflects our current responses and viewpoints. Thank you for your understanding and patience in this matter.

---

### Author Response · Authors · 2023-11-22
**Revision Summary for Reviewers**

Dear Reviewers,

We appreciate all your insightful feedback and have made several revisions to our paper to address the major concerns and enhance clarity. Our changes are **highlighted in blue**. Please refer to the latest version of our paper.

Below is a summary of the key changes:

1. **Revision of Equation (2):** We have refined the expression of Equation (2) and added a more detailed description. This change aims to clarify that Equation (2) represents a probability computation function of the pre-trained LM, reducing potential misunderstandings.

2. **Corrections to Equations (5) and (6):** We have corrected typos in these equations and relocated them to Appendix B.2. Additionally, we have expanded the explanation in the appendix to provide a more comprehensive understanding of these equations.

3. **Variable and Input-Output Dimensions:** We have specified the dimensions of each variable and detailed the input-output dimensions in the relevant sections. This enhancement is intended to improve the technical clarity of our model's architecture and functioning.

4. **Details on Node-type:** More information about node-type has been added to Appendix B.3. This addition provides how the node-type is determined and how it is used in the model.

5. **Rewriting of Section 3.3:** To reduce symbol overload and enhance readability, we have rewritten Section 3.3. We moved the (very) detailed and technical information to Appendix B.3. This restructuring aims to simplify the main text, making it more easy for readers to follow.

6. **Revisions to Algorithms 1 & 2:** Following the simplification of notation and changes in Section 3.3, we have updated the corresponding formulas in Algorithms 1 and 2. These revisions ensure consistency and clarity across the paper.

These changes are intended to enhance the technical clarity, readability, and overall quality of our paper. We believe that these revisions address your concerns effectively and make our research more understandable to the readers. Due to time constraints, we were unable to revise the paper further. However, we will continue to improve our work and address your feedback in the future versions.

Thank you for your valuable feedback, and we look forward to hearing from you.

\
Sincerely,

Authors